# Revisiting Group Relative Policy Optimization: Insights into On-Policy and Off-Policy Training

**Youssef Mroueh, Nicolas Dupuis, Brian Belgodere, Apoorva Nitsure, Mattia Rigotti, Kristjan Greenewald, Jiri Navratil, Jerret Ross & Jesus Rios** *

IBM Research

## Abstract

We revisit Group Relative Policy Optimization (GRPO) in both on-policy and off-policy optimization regimes. Our motivation comes from recent work on off-policy Proximal Policy Optimization (PPO), which improves training stability, sampling efficiency, and memory usage. In addition, a recent analysis of GRPO suggests that estimating the advantage function with off-policy samples could be beneficial. Building on these observations, we adapt GRPO to the off-policy setting. We show that both on-policy and off-policy GRPO objectives yield an improvement in the reward. This result motivates the use of clipped surrogate objectives in the off-policy version of GRPO. We then compare the empirical performance of reinforcement learning with verifiable rewards in post-training using both GRPO variants. Our results show that off-policy GRPO either significantly outperforms or performs on par with its on-policy counterpart.

## 1 Introduction

Proximal Policy Optimization (PPO) (Schulman et al., 2015; 2017) is a widely used algorithm in reinforcement learning. Reinforcement learning from Human Feedback (Christiano et al., 2017; Stiennon et al., 2020; Ouyang et al., 2022; Bai et al., 2022) and Reinforcement Learning from Verifiable Rewards (Lambert et al., 2024; Shao et al., 2024) are cornerstones in post-training of large language models to align their preferences with human values and to enable reasoning and coding capabilities using verifiable rewards.

Group Relative Policy Optimization introduced in (Shao et al., 2024) alleviates the need of training a critic network in PPO and uses Monte-Carlo samples referred to as "a group" to estimate the advantage function via a standardized reward, where the mean and standard deviation statistics are estimated using the group. GRPO was used to train the Deepseek R1 reasoning models (Guo et al., 2025) and was adopted by the open-source community as a method of choice for post-training of large language models, with open-source implementations in several libraries such as TRL of HuggingFace (von Werra et al., 2020b) and VERL (Luo et al., 2025).

Several recent works analyzed the loss implemented in GRPO such as Vojnovic and Yun (2025); Mroueh (2025). The study in Mroueh (2025) suggests that the iterative GRPO of Shao et al. (2024) with sample reuse (i.e. for $\mu > 1$ in Shao et al. (2024)) leads to an off-policy estimation of the advantage and to a success rate amplification when using verifiable rewards. Indeed, it has been observed empirically that this off-policy advantage estimation leads to an improved performance (HuggingFace, 2025b).

Motivated by these observations and the rich literature on off-policy PPO and RL, like work by Queeney et al. (2021); Meng et al. (2023); Gan et al. (2024); Fakoor et al. (2020) to cite a few (see related work Section 4 for a larger account on this), in this paper we explore the extension of GRPO to the off-policy regime where the advantage is estimated using statistics coming from a different policy than the current policy.

---

*Correspondence to: mroueh@us.ibm.com

The main contributions of this paper are:

- We review in Section 2 the iterative GRPO algorithm proposed in Shao et al. (2024) and introduce in Section 3 the off-policy GRPO.

- We show in Section 3 that the on-policy and off-policy advantages provide a lower bound on the policy improvement of the expected reward (Theorem 1 and Corollary 1).

- We state conditions under which optimizing the advantage leads to improvements in the off-policy regime, namely, given that the off-policy stays in the vicinity of the current policy and the variance of the reward under the off-policy is non zero, maximizing the regularized off-policy advantage leads to policy improvement. The regularization ensures that the updated policy stays close to the off-policy.

- Finally, armed with these results, we state the constrained policy optimization problem for off-policy GRPO in Section 3.2 and derive a clipped surrogate similar to the ones in off-policy PPO (Gan et al., 2024) and obtain on-policy GRPO clipped objective as a particular case.

- We validate experimentally that training LLMs with off-policy GRPO leads to either improved or on par performance while potentially reducing the communication burden in serving the model in each iteration for inference.

## 2 ON-POLICY GRPO

Let $\mathcal{X}$ be the space of inputs (prompts in the context of LLMs) and $\mathcal{Y}$ the space of responses. We denote by $\rho_{\mathcal{X}}$ the distribution on inputs. We refer to the policy we want to optimize as $\pi(\cdot|x)$, which is a distribution on $\mathcal{Y}$ conditioned on $x \sim \rho_{\mathcal{X}}$. For $k \geq 0$, let $\pi_k$ be the policy at the current step $k$.

The Group Relative Policy Optimization (GRPO) Clipped objective introduced in Shao et al. (2024) is a variant of Proximal Policy Optimization (PPO) (Schulman et al., 2017; 2015), where the advantage is computed as a standardized reward function with mean and variances computed with respect to a group or Monte-Carlo samples of size $G$ sampled from the current policy $\pi_k(.|x)$ for each $x$ independently. For $\epsilon, \beta > 0$ and given a reference policy $\pi_{\text{ref}}$, the clipped objective optimization in GRPO is defined as follows:

$$\max_{\pi} \mathbb{E}_{y \sim \pi_k(\cdot|x)} \min \left( \frac{\pi(y|x)}{\pi_k(y|x)} A_{\pi_k}(x,y), \ \text{clip} \left( \frac{\pi(y|x)}{\pi_k(y|x)}, 1-\epsilon, 1+\epsilon \right) A_{\pi_k}(x,y) \right) - \beta \text{KL}(\pi||\pi_{\text{ref}}),$$

where KL is the Kullback-Leibler divergence, and $A_{\pi_k}$ is the GRPO advantage function:

$$A_{\pi_k}(x,y) = \frac{r(x,y) - \mathbb{E}_{\pi_k} r(x,y)}{\sqrt{\mathbb{E}_{\pi_k}(r(x,y) - \mathbb{E}_{\pi_k} r(x,y))^2 + \varepsilon}}.$$

The advantage can be estimated from samples on "a group" of size $G$ for each $x$, we sample $y_1, \ldots, y_G \sim \pi_k(\cdot|x)$ and compute $r_\ell = r(x, y_\ell), \ell = 1, \ldots, G$. We refer to the group of reward conditioned on $x$ as $\{r_\ell\}$ and the estimated GRPO advantage is therefore (Shao et al., 2024):

$$\hat{A}_{\pi_k}(x, y_i) = \frac{r_i - \texttt{mean}(\{r_\ell\})}{\sqrt{\texttt{std}^2(\{r_\ell\}) + \varepsilon}},$$

where mean and std are empirical mean and standard deviation respectively. The statistics used to normalize the reward leading to the advantage function are estimated using the current policy $\pi_k$, and hence we refer to $A_{\pi_k}$ as the *on-policy advantage*.

When compared with PPO, GRPO alleviates the need of training a critic network to compute the advantage and relies instead on standardized rewards that can be estimated efficiently using efficient inference frameworks such as vLLM (Kwon et al., 2023) in the context of large language models.

**GRPO with Verifiable Rewards and Success Rate Amplification** The iterative GRPO (Shao et al., 2024) has two overlooked features:

- The algorithm suggests to optimize the policy $\pi$ for $\mu$ iterations fixing the samples from $\pi_k$, which inherently leads to an off-policy estimation of the advantage.

- The algorithm suggests to do the training in stages while changing $\pi_{\text{ref}}$ to the latest optimized policy with GRPO.

A recent analysis of GRPO with verifiable rewards, i.e. with binary rewards (Mroueh, 2025), suggests that this aforementioned off-policy advantage estimation in Shao et al. (2024) leads to an implicit fixed point iteration that guarantees that the success rate of the GRPO-optimized policy is higher than the one of the reference policy. This also explains the multi-stage nature of the iterative GRPO that changes the reference along the training iterations. Motivated by these observations, we propose to take a step back and analyze on-policy and off-policy GRPO. In practice, in our proposed off-policy GRPO instead of just fixing the samples for $\mu$ iterations from $\pi_k$ as suggested in Shao et al. (2024), we use the policy $\pi_{k-\mu}$ to estimate the advantage for $\mu$ iterations with fresh samples in each iteration, and we refer to this as *off-policy* advantage.

## 3 OFF-POLICY AND ON-POLICY GRPO REWARD IMPROVEMENT

We introduce in this Section off-policy GRPO, and analyze conditions under which policy reward improvement is possible in both the on-policy and off-policy regimes. Towards that goal, we start with some preliminary definitions. Define the expected reward of a policy given $x \sim \rho_{\mathcal{X}}$:

$$J(\pi(\cdot|x)) = \mathbb{E}_{y \sim \pi(\cdot|x)} r(x, y) \tag{1}$$

For $k \geq 0$, let $\pi_k$ be the policy at the current step $k$ and $\alpha(\cdot|x)$ be a policy used for off-policy sampling, where typically we consider $\alpha(\cdot|x) = \pi_{k-v}(\cdot|x)$, for $0 \leq v < k$. [1]

Define the mean and standard deviation of the off-policy reward, i.e. under policy $\alpha$: $\mu_{\alpha,r}(x) = \mathbb{E}_{y \sim \alpha(\cdot|x)} r(x, y)$ and $\sigma_{\alpha,r}(x) = \sqrt{\mathbb{E}_{y \sim \alpha(\cdot|x)} (r(x, y) - \mu_{\alpha,r}(x))^2}$, and denote for $0 < \varepsilon < 1$: $\sigma_{\alpha,r,\varepsilon}(x) = \sqrt{\sigma_{\alpha,r}^2(x) + \varepsilon}$.

The GRPO advantage function computed using the off-policy distribution $\alpha$ is defined as the whitened reward, as follows:

$$A_\alpha(x, y) = \frac{r(x, y) - \mu_{\alpha,r}(x)}{\sigma_{\alpha,r,\varepsilon}(x)}. \tag{2}$$

Our goal is to maximize the expected advantage function using importance sampling under the policy $\alpha$:

$$\mathcal{L}_\alpha(\pi(\cdot|x)) = \mathbb{E}_{y \sim \alpha(\cdot|x)} \frac{\pi(y|x)}{\alpha(y|x)} A_\alpha(x, y) \tag{3}$$

If $\alpha = \pi_k$, we obtain the online policy objective function of GRPO, where the advantage is computed with the current policy $\pi_k$, i.e. using $A_{\pi_k}(x, y)$.

### 3.1 POLICY IMPROVEMENT IN GRPO

Note that our goal is to optimize the expected reward under $\pi$, $J(\pi)$ given in eq. (1), but instead we use the expected advantage $\mathcal{L}_\alpha(\pi)$ – where the advantage is computed using $\alpha$ – given in eq. (3). Hence, our goal in what follows is to provide a lower bound on $J(\pi(\cdot|x)) - J(\pi_k(\cdot|x))$ that involves $\mathcal{L}_\alpha(\pi)$, which guarantees that maximizing the expected advantage function leads to improvement in terms of expected rewards on the current policy $\pi_k$.

Our lower bounds are given in Theorem 1 and Corollary 1 and they involve the total variation distance $\mathbb{TV}$ defined as follows:

$$\mathbb{TV}(m_1, m_2) = \frac{1}{2} \int |m_1 - m_2|.$$

**Theorem 1** (Policy Improvement Lower Bound in **Off-Policy GRPO**). *Assume that the reward is positive and bounded in $0 \leq r \leq 1$. Let $\alpha$ be the off-policy distribution and $\pi_k$ the current policy. Then for any policy $\pi$ we have for all $x$ ($\rho_{\mathcal{X}}$ a.s.):*

---

[1] Note in Section 2 we referred to this as $\pi_{k-\mu}$ so we keep close to notation used in the original GRPO paper. We will use $v$ instead of $\mu$ in the rest of the paper.

$$J(\pi(\cdot|x)) - J(\pi_k(\cdot|x)) \geq \mathcal{L}_\alpha(\pi(\cdot|x)) - 2\frac{1 - \sigma_{\alpha,r,\varepsilon}(x)}{\sigma_{\alpha,r,\varepsilon}(x)} \mathbb{TV}(\pi(\cdot|x), \alpha(\cdot|x)) - 2\, \mathbb{TV}(\pi_k(\cdot|x), \alpha(\cdot|x))$$

If the reward is not bounded by 1 we can scale it by $\|r\|_\infty$ so it becomes in $[0,1]$, without this impacting the overall optimization problem (for a negligble $\varepsilon$). Note that this condition on the reward ensures that $\sigma_{\alpha,r}(x) \leq 1$ which is needed in the GRPO case to get the policy improvement lower bound. Indeed for bounded random variable in $[a,b]$ the variance is bounded by $\frac{(b-a)^2}{4}$, and hence we have $\sigma_{\alpha,r}(x) \leq \frac{1}{2}$, which guarantees that the term $\frac{1-\sigma_{\alpha,r,\varepsilon}(x)}{\sigma_{\alpha,r,\varepsilon}(x)} \geq 0$ , for $\varepsilon \leq \frac{3}{4}$. $(\sigma_{\alpha,r,\varepsilon}(x) \leq \sqrt{\frac{1}{4} + \varepsilon} \leq 1)$ .

For on-policy GRPO i.e. setting $\alpha = \pi_k$ in Theorem 1 we have the following corollary:

**Corollary 1** (Policy Improvement Lower Bound in **On-Policy GRPO**). *Assume that the reward is positive and bounded, $0 \leq r \leq 1$, and consider (negligible)$\varepsilon$ such that $0 < \varepsilon \leq \frac{3}{4}$. Let $\pi_k$ be the current policy, then for any policy $\pi$ we have for all $x$ ($\rho_\mathcal{X}$ a.s.):*

$$J(\pi(\cdot|x)) - J(\pi_k(\cdot|x)) \geq \mathcal{L}_{\pi_k}(\pi(\cdot|x)) - 2\frac{1 - \sigma_{\pi_k,r,\varepsilon}(x)}{\sigma_{\pi_k,r,\varepsilon}(x)} \mathbb{TV}(\pi(\cdot|x), \pi_k(\cdot|x))$$

Define :

$$M_{\alpha,r,\varepsilon} = \sqrt{\mathbb{E}_{x \sim \rho_\mathcal{X}} \frac{(1 - \sigma_{\alpha,r,\varepsilon}(x))^2}{\sigma_{\alpha,r,\varepsilon}^2(x)}}$$

Integrating Theorem 1 on $x$ (prompts) and applying Cauchy-Schwarz inequality we obtain:

$$\mathbb{E}_{x \sim \rho_\mathcal{X}} J(\pi(\cdot|x)) - \mathbb{E}_{x \sim \rho_\mathcal{X}} J(\pi_k(\cdot|x)) \geq \mathbb{E}_{x \sim \rho_\mathcal{X}} \mathcal{L}_\alpha(\pi(\cdot|x))..$$
$$.. - 2M_{\alpha,r,\varepsilon}(\mathbb{E}_{x \sim \rho_\mathcal{X}} \mathbb{TV}^2(\pi(\cdot|x), \alpha(\cdot|x)))^{\frac{1}{2}} - 2\mathbb{E}_{x \sim \rho_\mathcal{X}} \mathbb{TV}(\pi_k(\cdot|x), \alpha(\cdot|x)) \qquad (4)$$

**Interpreting the lower bound** When compared with lower bounds for policy improvement in PPO (Theorem 1 in TRPO (Schulman et al., 2015)) and for off-policy PPO (Lemma 3.1 in transductive PPO (Gan et al., 2024) and Theorem 1 in Generalized PPO (Queeney et al., 2021)), we observe similar lower bounds with a crucial difference that the constants weighting total variations are absolute constants for PPO whereas they are policy and data dependent for GRPO. In particular, the dependency of the lower bound on:

$$\frac{1 - \sigma_{\alpha,r,\varepsilon}(x)}{\sigma_{\alpha,r,\varepsilon}(x)}$$

is of interest. We can examine this quantity for verifiable rewards, for each $x$ the verifiable reward is a Bernoulli random variable with parameter $p$ the probability of success of the policy given $x$ (Mroueh, 2025). Hence we have:

$$\frac{1 - \sigma_{\alpha,r,\varepsilon}(x)}{\sigma_{\alpha,r,\varepsilon}(x)} = \frac{1 - \sqrt{p(1-p) + \varepsilon}}{\sqrt{p(1-p) + \varepsilon}}$$

We observe that this quantity diverges for fully correct and incorrect answers and this can indeed hurt the lower bound, as the negative terms in the lower bound will be dominating. It was suggested in DAPO (Yu et al., 2025) to filter out prompts with fully correct or incorrect answers, this will have the effect of controlling this term in the lower bound and keep that quantity finite.

**Comparison To Off-Policy PPO Improvement Lower Bounds.** While the statements of the theorems are similar to off-policy PPO algorithms (GePPO/ToPPO) ((Gan et al., 2024) and(Queeney et al., 2021)) they crucially follow different proof techniques. The main innovation is in extending these results to GRPO. The proof in (Gan et al., 2024) uses the results from (Kakade and Langford, 2002) and (Achiam et al., 2017) , in order to establish the lower bound. In particular, the proof uses the result from (Kakade and Langford, 2002) to link the difference of expected rewards between

two different policies to the total variation between the state visitation distributions, and relies on (Achiam et al., 2017) to link this quantity to the total variation between the policies. Note that this reliance on the state visitation distributions restricts the analysis to a Markov decision Process (MDP) setup. Our proof technique exploits the analytical form of the GRPO advantage and does not need the state visitation distribution usually needed in PPO analysis (online and offline). This makes our analysis more general and does not need to be in the MDP setup.

## 3.2 GRPO: FROM CONSTRAINED OPTIMIZATION TO CLIPPED SURROGATE OBJECTIVES

**From Penalized to KL Constrained Optimization** To maximize the lower bound in eq.(4), we see that the off-policy $\alpha$ needs to be in the vicinity of the current policy $\pi_k$, i.e. for $\mathbb{TV}(\alpha, \pi_k) \leq \delta$ and that $\boxed{M_{\alpha,r,0} < \infty}$ (variance terms not exploding). Under these assumptions, we can solve the following penalized problem :

$$\max_{\pi} \mathbb{E}_{x \sim \rho_{\mathcal{X}}} \mathcal{L}_\alpha(\pi(\cdot|x)) - 2\, M_{\alpha,r,\varepsilon} \sqrt{\mathbb{E}_{x \sim \rho_{\mathcal{X}}} \mathbb{TV}^2(\pi(\cdot|x), \alpha(\cdot|x))}.$$

By virtue of Theorem 1, maximizing this objective above leads to policy reward improvement.

We can write this as a constrained optimization, there exists $\Delta > 0$ such that the following constrained optimization problem is equivalent:

$$\max_{\pi} \mathbb{E}_{x \sim \rho_{\mathcal{X}}} \mathcal{L}_\alpha(\pi(\cdot|x)) \text{ subject to } \mathbb{E}_{x \sim \rho_{\mathcal{X}}} \mathbb{TV}^2(\pi(\cdot|x), \alpha(\cdot|x)) \leq \Delta^2.$$

By Pinsker inequality for two measures $m_1, m_2$ we have $\mathbb{TV}(m_1, m_2) \leq \sqrt{\frac{1}{2}\mathsf{KL}(m_1, m_2)}$ and hence we can bound instead the KL divergence as follows:

$$\boxed{\max_{\pi} \mathbb{E}_{x \sim \rho_{\mathcal{X}}} \mathcal{L}_\alpha(\pi(\cdot|x)) \text{ subject to } \frac{1}{2} \mathbb{E}_{x \sim \rho_{\mathcal{X}}} \mathsf{KL}(\pi(\cdot|x), \alpha(\cdot|x)) \leq \Delta^2.} \quad (5)$$

**From Constrained Optimization to Clipped Surrogate Objectives** The objective in (5) is the same as in the original constrained PPO formulation (Schulman et al., 2015) with two key differences: the advantage is the whitened reward of GRPO where the statistics are computed using the off-policy $\alpha$ , and the advantage objective is computed using importance sampling from the off-policy $\alpha$, instead of $\pi_k$ in both cases. This is indeed related to objectives in off-policy PPO (Queeney et al., 2021; Gan et al., 2024). A practical implementation of these objectives is through clipped surrogates (Schulman et al., 2015).

For $\epsilon \in [0, 1]$ following Gan et al. (2024); Queeney et al. (2021) let us define:

$$f_\epsilon(r, r', a) = \min(ra,\ \text{clip}(r, \max(r' - \epsilon, 0), r' + \epsilon)\, a).$$

The clipped off-policy GRPO objective for $\alpha$ such that $\boxed{\mathbb{TV}(\alpha, \pi_k) \leq \delta}$ and $\boxed{M_{\alpha,r,0} < \infty}$ is therefore defined as follows : $\mathcal{L}_\alpha^c(\pi(\cdot|x)) = \mathbb{E}_{y \sim \alpha(\cdot|x)} f_\epsilon\left(\frac{\pi(y|x)}{\alpha(y|x)}, \frac{\pi_k(y|x)}{\alpha(y|x)}, A_\alpha(x, y)\right)$

Let us unpack this, we have: $f_\epsilon\left(\frac{\pi(y|x)}{\alpha(y|x)}, \frac{\pi_k(y|x)}{\alpha(y|x)}, A_\alpha(x, y)\right) = ..$

$$.. = \begin{cases} A_\alpha(x, y) \min\left(\frac{\pi(y|x)}{\alpha(y|x)}, \frac{\pi_k(y|x)}{\alpha(y|x)} + \epsilon\right), & r(x, y) \geq \mu_{\alpha,r}(x) \\ A_\alpha(x, y) \max\left(\frac{\pi(y|x)}{\alpha(y|x)}, \max(\frac{\pi_k(y|x)}{\alpha(y|x)} - \epsilon, 0)\right), & r(x, y) < \mu_{\alpha,r}(x). \end{cases}$$

The clipping ensures that the ratio $\frac{\pi}{\alpha}$ remains bounded and is a relaxation of the KL (or the total variation distance). Since $\alpha$ needs to satisfy closeness to $\pi_k$ in order to ensure improvement, the clipping objective incentivizes the difference between $\frac{\pi}{\alpha} - \frac{\pi_k}{\alpha}$ to not exceed $\epsilon$ (Gan et al., 2024).

In practice, the off-policy is $\alpha = \pi_{k-v}$ for a small $v \in [0, k)$. Given a small learning rate and a small $v$, the assumption that the policy $\pi_{k-v}$ doesn't deviate from $\pi_k$ is reasonable, and for $v$ small we can approximate $\frac{\pi_k}{\pi_{k-v}}$ by 1. We use this approximation in practice as we found it more stable, and

given that this approximation is in practice used in off-Policy PPO (with sample reuse) as discussed in Gan et al. (2024) (See Section 4.1 in Gan et al. (2024)).

**Back to On-Policy GRPO Clipped Objective** For $\alpha = \pi_k$, we obtain the clipped objective for on-policy GRPO (Shao et al., 2024):

$$
\mathcal{L}^c_{\pi_k}(\pi(\cdot|x)) = \mathbb{E}_{y \sim \pi_k(\cdot|x)} f_\epsilon \left( \frac{\pi(y|x)}{\pi_k(y|x)}, 1, A_{\pi_k}(x,y) \right)
$$

$$
= \mathbb{E}_{y \sim \pi_k(\cdot|x)} \min \left( \frac{\pi(y|x)}{\pi_k(y|x)} A_{\pi_k}(x,y), \mathrm{clip} \left( \frac{\pi(y|x)}{\pi_k(y|x)}, 1-\epsilon, 1+\epsilon \right) A_{\pi_k}(x,y) \right).
$$

$\mathsf{KL}-$ **Regularized RL & On-Policy / Off-Policy Algorithms** Finally putting together our clipped surrogate objective with the KL regularizer we obtain our final objective:

$$
\mathbb{E}_{x \sim \rho_{\mathcal{X}}} \mathcal{L}^c_\alpha(\pi(\cdot|x)) - \beta \mathsf{KL}(\pi(\cdot|x)||\pi_{\mathrm{ref}(\cdot|x)}). \tag{6}
$$

We present the GRPO algorithm in Algorithm 1 and the configurations that allow toggling between on-policy and off-policy GRPO in Table 1. Within the RL loop, the model is served for inference using vLLM (Kwon et al., 2023). The parameter $v$ controls how often the model is updated on the vLLM server (which corresponds to off-policy with $\alpha = \pi_{k-v+1}$). The parameter $i$ controls how many SGD iterations are applied to each batch sampled from the policy. For $v = 1$ and $i = 1$, the model is continuously served, and each batch of samples is used once in SGD. This corresponds to on-policy GRPO. For $i > 1$ and $v = 1$, the model is still continuously served, but each batch is used $i$ times in the SGD loop; this corresponds to an "off-policy" GRPO variant, as proposed in Shao et al. (2024). For large models that require tensor parallelism and multi-GPU serving, continuous model serving incurs additional communication costs. Our off-policy GRPO mitigates these costs by serving the model every $v > 1$ iterations (line 8 in Algorithm 1) and fixing $i = 1$. Our theory guarantees reward improvement as long as $v$ is not too large.

**Computational and Communication Costs** Updating the model served by vLLM during GRPO training incurs varying costs depending on the model size, update frequency ($v$), and parallelism settings. When the training model and vLLM instance reside on different GPUs, or when vLLM uses tensor parallelism (TP), model updates may trigger deep copies and inter-GPU communication. These involve either full weight transfers or partitioned broadcasts, which scale linearly with model size. Frequent updates (e.g., $v = 1$) can dominate the runtime, especially for large models (see the recent benchmark vLLM (2025) for latencies in serving large models with tensor parallelism using vLLM). To mitigate this, we update the vLLM model every $v > 1$ iterations. This amortizes the copy cost while maintaining reward improvement guarantees from our theory. For instance, in GRPO's TRL parallel implementation (von Werra et al., 2020b), there are two modes: a single-server dedicated for inference and a co-located training and inference (Toslali et al., 2025). For instance, in GRPO's TRL parallel implementation (von Werra et al., 2020b), there is a server and a colocate implementation (Toslali et al., 2025). While the single vLLM server assigns a single GPU for serving the model on a single node, colocate shards the vLLM inference model using tensor parallelism (TP) on all available GPUs, colocating the inference and the training on all GPUs. In colocate mode, GRPO can benefit from off-policy updates, as the model does not need to be sharded and served at each iteration of the training when $v > 1$.

**On-Policy Clipped Objective with Zero Variance Masking a la DAPO (Yu et al., 2025)** As discussed earlier in the interpretation of the lower bound in page 4, the samples with zero variance may lead to total variation terms to dominate the lower bound, hence we propose similar to DAPO (Yu et al., 2025) to mask these samples. For instance in the on policy case this would be with the following masked objective:

$$
\mathbb{E}_{x \sim \rho_{\mathcal{X}}} \mathbb{1}_{\sigma_{\pi_k,r}(x) \neq 0} \left( \mathcal{L}^c_{\pi_k}(\pi(\cdot|x)) - \beta \mathsf{KL}(\pi(\cdot|x)||\pi_{\mathrm{ref}}(\cdot|x)) \right). \tag{7}
$$

**Remark 1.** *We make the following remarks on off-policy GRPO:*

- $(\mathbf{i} > \mathbf{1}, \mathbf{v} = \mathbf{1})$ *As Off-Policy : This setup was proposed in (Shao et al., 2024), and this form of data reuse can be traced back to PPO (Schulman et al., 2017). (Gan et al., 2024) analyzed this setup in PPO as an off-policy training regime (Section 4.1 in (Gan et al., 2024)), and we refer to it as off-policy following their lead.*

| Method name | Update by fixed batch $i$ | Update of Policy on Server $v$ |
|---|---|---|
| On-Policy GRPO (Shao et al., 2024) | $i = 1$ | $v = 1$ |
| Off-policy GRPO (Shao et al., 2024) | $i > 1$ | $v = 1$ |
| Off-policy GRPO (this work) | $i = 1$ | $v > 1$ |

Table 1: Training configurations in alg. 1: (v1-i1) is on-policy GRPO and (v1-i10) is an example of off-policy GRPO in (Shao et al., 2024). Our off-policy GRPO corresponds e.g. to (v10-i1).

- ***Zero-Variance Masking and Bias*** *In practice the variance is estimated using a finite group size G. Masking based on that estimate introduces a bias especially with small group size. (Yu et al., 2025) proposes a dynamic group size that makes sure that at least a success and failure exist within each group, this lowers the bias and caps the number of masked samples.*

---

**Algorithm 1** Iterative GRPO with verifiable rewards, modified from Shao et al. (2024)

---

1: **Input** initial policy model $\pi_{\theta_{\text{init}}}$; verifiable reward $r$; task prompts $\mathcal{D}$;
2: **Hyperparameters** $\epsilon$, $\beta$, $S$,
3: $(i, v)$=(Number of SGD iteration by fixed batch, Model update on vLLM server)
4: **Policy model** $\pi_\theta \leftarrow \pi_{\theta_{\text{init}}}$, $\pi_{\text{ref}} \leftarrow \pi_{\theta_{\text{init}}}$, $\pi_{\theta_{\text{old}}} \leftarrow \pi_\theta$
5: **for** $s = 1, \ldots, S$ **do**
6:     **for** $k = 1, \ldots, M$ **do**
7:         Sample a batch $\mathcal{D}_b$ from $\rho_{\mathcal{X}}$
8:         **if** $k \bmod v = 0$ **then**
9:             Update the old policy model on the vLLM server $\pi_{\theta_{\text{old}}} \leftarrow \pi_\theta$
10:         **end if**
11:         Sample $G$ outputs $\{y_i\}_{i=1}^G \sim \pi_{\theta_{\text{old}}}(\cdot \mid x_i)$ for each question $x \in \mathcal{D}_b$
12:         Compute rewards $\{r_i\}_{i=1}^G$ for each sampled output $y_i$ by running verifiable reward $r$
13:         $\alpha \leftarrow \pi_{\theta_{\text{old}}}$
14:         Compute $A_\alpha(x, y_i)$ using Equation (2)
15:         **for** GRPO iteration $= 1, \ldots, i$ **do**           $\triangleright$ $i$ is referred to as $\mu$ in Original GRPO
16:             Update the policy model $\pi_\theta$ by maximizing the GRPO objective (6) with gradient ascent
17:         **end for**
18:     **end for**
19:     $\pi_{\text{ref}} \leftarrow \pi_\theta$           $\triangleright$ Swap reference with the latest model
20: **end for**
21: **Output** $\pi_\theta$

---

## 4 RELATED WORK

**Proximal Policy Optimization (PPO) and Extensions** Proximal Policy Optimization (PPO) is a widely used on-policy reinforcement learning algorithm that improves training stability through clipped surrogate objectives. While PPO is effective in diverse settings, it is inherently limited by its on-policy nature, which constrains sample efficiency. To address these limitations, several off-policy adaptations and extensions of PPO have been proposed. Generalized Proximal Policy Optimization (G-PPO) (Queeney et al., 2021) enables sample reuse while maintaining convergence guarantees. Transductive off-Policy PPO (ToPPO) (Gan et al., 2024) builds on G-PPO by incorporating transductive learning principles, bridging the gap between off-policy learning and theoretical guarantees of on-policy methods. Off-Policy PPO (OPPO) (Meng et al., 2023) proposes novel corrections to integrate replay buffer samples in PPO-style updates.

**On-Policy and Off-Policy Actor-Critic Methods** Actor-critic methods blend the strengths of policy gradients and value function estimation. Off-policy variants aim to improve sample efficiency by learning from a replay buffer. The Off-Policy Actor-Critic algorithm (Degris et al., 2012) introduces importance weighting to enable stable updates from off-policy data. ACER (Wang et al., 2016) extends this with trust-region optimization and truncated importance sampling, enhancing by that the learning efficiency in discrete action spaces. Mixing on-policy and off-policy methods aims to leverage the stability of on-policy updates with the efficiency of off-policy learning. P3O (Fakoor

et al., 2020) provides a principled approach that interleaves policy updates from both on- and off-policy data.

**Off-Policy RLHF and other variants of GRPO** Noukhovitch et al. (2025) introduced within the iterative DPO framework an asynchronous RLHF using off-policy data and that ensures faster convergence to the optimal policy. New variants of GRPO have been proposed recently such as DAPO (Yu et al., 2025) and DR-GRPO (Liu et al., 2025). DAPO proposes the zero variance masking without theoretical backing, our work roots this in the improvement lower bound. DR-GRPO proposes to center only the reward without using the variance normalization.

## 5 EXPERIMENTS

### 5.1 ABLATION STUDIES ON GSM8K

**Setup, Model, and Data** In our first set of experiments, we use `GSM8K` dataset from Cobbe et al. (2021) (MIT license), and `Qwen/Qwen2.5-0.5B-Instruct` (Apache 2.0 license) by Yang et al. (2024). We integrate our changes in Algorithm 1 to the GRPO implementation in TRL (von Werra et al., 2020b), and train our models on the training split of `GSM8K` on a node with 8 GPUs ($GPU_0$ for the vLLM server and 7 other GPUs for distributed training). See Appendix A for the hardware specification. We use a learning rate of $5 \times 10^{-6}$ for all experiments and the KL regularizer $\beta = 0.1$ in Equation (6). We use the correctness of the LLM output as a reward. For GRPO training, the hyperparameters are the following: group size $G = 16$ and per-device batch size 16 (meaning each GPU processes a single prompt $x$ with 16 responses). To increase the overall batch-size we use gradient accumulation of 4, ending with an effective batch size of prompts of 28. The context length used for this experiment is 200, and the sampling temperature is set to $\tau = 0.1$.

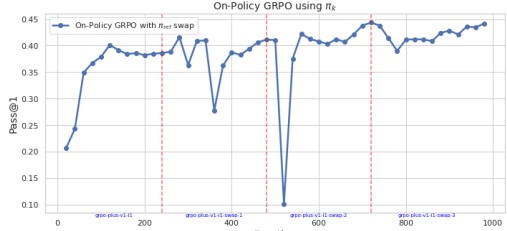

(a) On-Policy GRPO with $\pi_{\text{ref}}$ swap at end of each epoch. ($v = 1$, $i = 1$, $S = 3$)

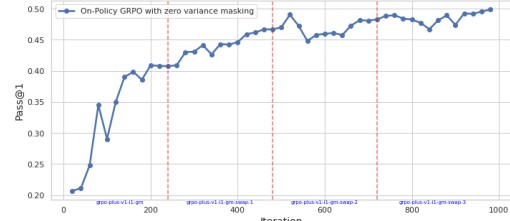

(b) On-Policy GRPO with masking of samples with variance $\sigma_{\pi_k,r} = 0$, and with $\pi_{\text{ref}}$ swap at end of each epoch. $v = 1$, $i = 1$, $S = 3$)

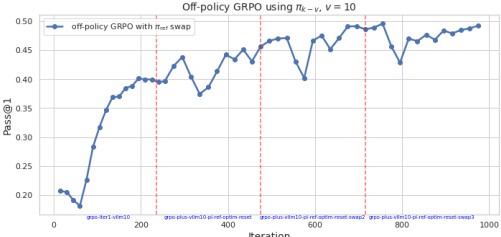

(c) Off-Policy GRPO using $v = 10$ (this amounts to fixing the model on the vLLM server for 10 iterations and getting fresh samples for new batches), and with $\pi_{\text{ref}}$ swap. ($v = 10$, $i = 1$, $S = 3$)

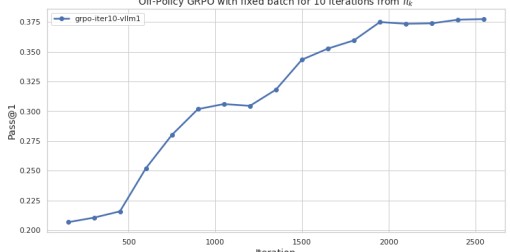

(d) Off-Policy GRPO using fixed samples from $\pi_k$ for 10 iterations. This will make 1 epoch $10\times$ slower. $v = 1$, $i = 10$, $S = 1$)

Figure 1: We train different variants of GRPO on the train portion of GSM8K and report the Pass@1 on GSM8K test set using 50 samples for each question in the test set for various variants of on-policy and off-policy GRPO. We see that as predicted by our theory, masking samples with zero variance stabilizes the training for on-policy training and leads to better performance. For off-policy training we see that using $v = 10$, $i = 1$ stabilizes also the training and leads also to better performance.

**Ablations and Results**   We train our models with GRPO using Algorithm 1 with a verifiable reward for answer correctness. We use for GRPO different configurations given in Table 1 and report on the test split of `GSM8K` Pass@1 using 50 samples (i.e. frequency of success given 50 generations for each question) using the same sampling configuration as in training. We report results in Figure 1: Fig. 1a for on-policy GRPO ($i = 1, v = 1$) with the objective given in Equation (6) with $S = 3$ (i.e. for 4 epochs with $\pi_{\text{ref}}$ swap at end of each epoch with latest model); Fig. 1b for on-policy GRPO ($i = 1, v = 1$) with masking zero variance samples i.e. using the objective given Equation (7) with $S = 3$; Fig. 1c for our off-policy GRPO ($v = 10, i = 1$), with $S = 3$ and Fig. 1d for Shao et al. (2024)'s off-policy GRPO i.e ($v = 1, i = 10$) for a single epoch. Note that we limit here our experimentation for ($v = 1, i = 10$) to $S = 1$ for fairness as this setup is the most computationally expensive, it is $10\times$ slower than all the other setups (($i = 1, v = 1$) and ($i = 1, v = 10$)).

We see in Fig. 1a that while the on-policy GRPO converges to a maximum Pass@1 of $45\%$ it is unstable. The masking of zero variance sampling in 1b stabilizes the on-policy GRPO and leads to an improvement of the performance to $50\%$. This is in line with our theoretical grounding through the improvement lower bound. Our off-policy GRPO in Fig. 1c stabilizes the training also and leads to an improved Pass@1 of $50\%$ on the test set. In all three cases, we see that by resetting the $\pi_{\text{ref}}$ to the latest model, GRPO amplifies the success rate above the current $\pi_{\text{ref}}$, this concurs with the theoretical findings in Mroueh (2025). Finally, the off-policy variant in Shao et al. (2024) in Fig. 1d shows a slower convergence over an epoch. Note that the standard deviation of Pass@1 over three seeds across iterations ranges between 0.003 and 0.03.

## 5.2   FINETUNING QWEN DISTILL R1 MODEL (1.5 B) ON DEEPSCALER DATA

In this section we use GRPO to finetune `DeepSeek-R1-Distill-Qwen-1.5B` (Guo et al., 2025) on `DeepScaleR-Preview-Dataset` from Luo et al. (2025) consisting of roughly $40K$ math questions with known answers. We used `math-verify` as the verifiable reward. We use a learning rate of $1 \times 10^{-6}$ in the same distributed setting as before (GPU$_0$ for vLLM and 7 GPUs for distributed training). We use a context length of 4096, a group size $G = 16$, a per-device batch size of 16, and the KL regularizer is $\beta = 0.001$. The sampling temperature used is $0.7$. We compared here the on-policy GRPO ($v = 1, i = 1$) to our off-policy GRPO ($v = 10, i = 1$) and report the performance of the trained model on a single epoch (around 24 hours on a single node). We report in Tables 2 and 3 Aime24 and Math500 performance using Huggingface LightEval (Habib et al., 2023). Aime24 is evaluated with Pass@1 using 32 samples, and math500 with extractive matching as recommended in lighteval with a context length of $32K$ (evaluation context length and all other sampling hyperparameters are set to the default in OpenR1 for this model). Plots of evaluation as function of iterations are given in Appendix D. We see that both on-policy and off-policy GRPO improve the performance of `DeepSeek-R1-Distill-Qwen-1.5B` that has an Aime24 of $29\%$ to $32\%$ at maximum (over iterations), and its math-500 from $83\%$ to $87\%$. This result confirms our theoretical results that by going off-policy we don't lose in terms of overall performance.

| Model/Aime24 | Min | Max | Median | Mean |
|---|---|---|---|---|
| v1-i1-length-4096 | 0.2802 | 0.3229 | 0.3021 | 0.3022 |
| v10-i1-length-4096 | 0.2781 | 0.3250 | 0.3047 | 0.3049 |

Table 2: Aime24 using lighteval with on & off-policy ( `(v1-i1)` and `(v10-i1)`) GRPO.

| Model/Math500 | Min | Max | Median | Mean |
|---|---|---|---|---|
| v1-i1-length-4096 | 0.830 | 0.870 | 0.854 | 0.8519 |
| v10-i1-length-4096 | 0.822 | 0.872 | 0.846 | 0.8474 |

Table 3: Math 500 extractive matching using lighteval (Habib et al., 2023) with on and off-policy `(v1-i1)` and `(v10-i1)` GRPO.

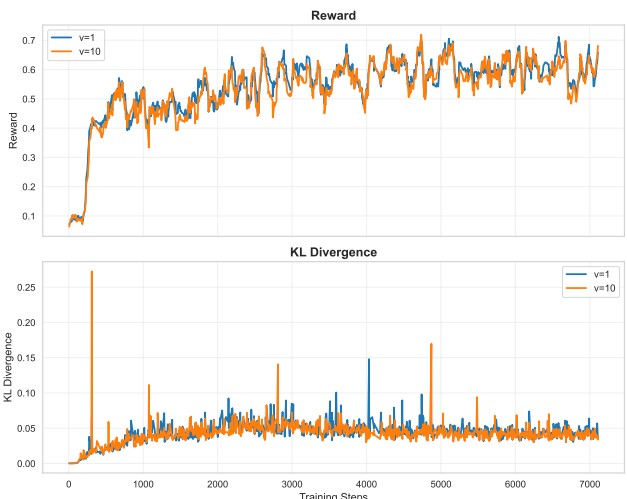

Figure 2: Reward and KL comparison of on-policy (blue) and off-policy (orange) GRPO with the Qwen2.5-7B model on the reasoning math dataset orz-math-57k.

## 5.3 SPEEDUP WITH A 7B MODEL AND COLOCATE GRPO USING TENSOR PARALLELISM

We ran scaling experiments on the Qwen2.5-7B model on the reasoning math dataset orz-math-57k (Hu et al., 2025). We used the new implementation of GRPO in TRL with colocate (Toslali et al., 2025) that colocates training and inference using tensor parallelism. On a single node with 8 GPUs, we trained using trl's GRPO with vLLM- colocate and a tensor parallel size TP = 4. This means that we have two vLLM servers sharded on 4 GPUs each. In this setup inference on the two vLLM servers is colocated with the training with 70% memory for training and 30% for vLLM. We used $\beta = 0$, a max completion length of 4096, a per-device-batch-size of 4, and a group size for GRPO of 8, and maximum training steps of 400.

We baselined the on-policy GRPO (v=1) and the off -policy GRPO (v=10) within this setup. We observed for on-policy (v=1) $0.02828 \pm 0.017$ iterations/sec and for off-policy (v=10) $0.03828 \pm 0.016$ iterations/sec. In this experiment with a 7 B model and tensor parallelism (TP=4) on a single node we therefore observe **1.35** $\times$ speedup for the off-policy with respect to the on-policy GRPO. Both on-policy and off-policy training result in the same reward on the orz-math-57k (50%), this supports our claims that off-policy does not compromise the training but leads to reduction in wall clock and in memory traffic due to serving the model. We give in Fig. 2 a comparison of the on-policy and off-policy GRPO in terms of the reward and the KL of the current policy to the reference model as function of the training iteration. We see that while the KL spikes at certain iterations in the off-policy case (`v10-i1`), this does not undermine the performance that tracks well on-policy GRPO (`v1-i1`).

## 6 CONCLUSION AND DISCUSSION

We revisited (on-policy) GRPO (Shao et al., 2024) and showed that its clipping objective can be derived from first principles as a lower bound for reward improvement. We also gave theoretical grounding to masking of zero variance samples suggested in DAPO (Yu et al., 2025). We introduced off-policy GRPO and laid out conditions under which it leads to policy improvement. Our off-policy GRPO has the advantage of reducing communication costs in serving the model for inference within the GRPO loop at each iteration as done in the on-policy counterpart, while not sacrificing performance. We showcased that off-policy GRPO stabilizes training and leads to either on par or improved performance as the on-policy one. The main takeaways of our paper to practitioners are: (1) Zero variance masking stabilizes on-policy GRPO's training (2) Off-policy GRPO attains its full potential in terms of maintaining performance and lowering latencies and communication overhead in larger scale training where models are served using tensor parallelism (see vLLM (2025)).

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

REPRODUCIBILITY

We provide the code needed to reproduce all of our experiments in supplement.

BROADER IMPACT AND LIMITATIONS

Our work analyzes the celebrated GRPO algorithm and develops an adaptation for the off-policy setting motivated by recent efforts for PPO that demonstrated higher stability and efficiency. Our primary contributions are theoretical, providing formal conditions under which advantage optimization guarantees policy improvement for the on-policy and off-policy regimes. These insights provide lower bounds on policy improvement and directly inform a practical clipped surrogate optimization objective for large language model (LLM) policy training that inherits our theoretical guarantees for both on policy and off policy regimes. In the on-policy regime our lower bound sheds light and gives theoretical backing to the benefits of masking samples with zero variance as suggested in the DAPO paper (Yu et al., 2025). Our formulation also clarifies theoretical relationships between our newly introduced off-policy GRPO, PPO variants, and general off-policy optimization frameworks – a linkage previously underexplored in the literature. Our derived off-policy GRPO algorithm is validated experimentally demonstrating improved performance compared to standard GRPO, while having the potential to reduce the communication overhead across devices in serving large models for sampling that is needed in GRPO. The broader impacts that we anticipate from our work (besides those directly inherited from GRPO and reinforcement fine-tuning of LLMs and the risks associated to the dual use of the enabled reasoning models) are then generally positive, as it enhances RL efficiency, reducing computational costs and improving stability.

## A ASSETS

**Hardware setup** All our experiments were run on one compute node with Dual 48-core Intel Xeon 8468, 2TB of RAM, 8 NVIDIA HGX H100 80GB SXM5, 8x 3.4TB Enterprise NVMe U.2 Gen4, and 10x NVIDIA Mellanox Infiniband Single port NDR adapters, running RedHat Enterprise Linux 9.5

**Libraries** Our experiments rely on the open-source libraries `PyTorch` (Paszke et al., 2019) (license: BSD), `HuggingFace Transformers` (Wolf et al., 2020) (Apache 2.0 license), and `HuggingFace TRL` (von Werra et al., 2020a) (Apache 2.0 license). We also relied on Open-R1 (HuggingFace, 2025a) as well as LightEval (Habib et al., 2023) for the evaluation of Aime24 and Math500.

**Code re-use** Our GRPO training code is based on the public Github repository `https://github.com/huggingface/open-r1` (HuggingFace, 2025a).

**Data and Models** In our experiments, we use the following publicly available datasets: (1) GSM8K dataset from Cobbe et al. (2021) (MIT license), and (2) the `DeepScaleR-Preview-Datasets` from Luo et al. (2025) (MIT license). The models that we used were `Qwen/Qwen2.5-0.5B-Instruct` (Apache 2.0 license) by Yang et al. (2024), and `DeepSeek-R1-Distill-Qwen-1.5B` (MIT license) by Guo et al. (2025).

## B TERMINOLOGY USED IN THE PAPER

- **GSM8K:** A standard grade-school mathematics reasoning benchmark consisting of short word problems with verifiable numeric answers.
- **DeepScaleR-Preview-Datasets:** A collection of math and reasoning datasets curated for verifiable reward modeling and fine-tuning of reasoning-oriented LLMs.
- **AIME24:** The 2024 American Invitational Mathematics Examination benchmark, used to assess advanced problem-solving and reasoning capabilities.
- **Extractive matching:** Automatic evaluation that parses and matches numeric or symbolic answers from model outputs against ground-truth solutions.

- **LightEval:** An open-source evaluation toolkit for LLMs supporting standardized benchmarking, prompt formatting, and metric computation.
- **Pass@1:** The accuracy metric representing the fraction of problems solved correctly by the first generated sample.

## C    REWARD IMPROVEMENT LOWER BOUND

### C.1    PROOFS OF THEOREM 1

We have :

$$J(\pi(\cdot|x)) = \mathbb{E}_{y\sim\pi(\cdot|x)} r(x,y)$$

Let $\pi_k$ be the current policy and $\alpha(\cdot|x)$ be another policy typically consider $\alpha(\cdot|x) = \pi_{k-i}(\cdot|x)$.

Define mean and variances of the off-policy reward, i.e policy under $\alpha$:

$\mu_{\alpha,r}(x) = \mathbb{E}_{y\sim\alpha(\cdot|x)} r(x,y)$ and $\sigma_{\alpha,r}(x) = \sqrt{\mathbb{E}_{y\sim\alpha(\cdot|x)}(r(x,y) - \mu_{\alpha,r}(x))^2}$, and denote for $0 < \varepsilon < \frac{3}{4}$: $\sigma_{\alpha,r,\varepsilon}(x) = \sqrt{\sigma_{\alpha,r}^2(x) + \varepsilon}$.

Note that we have a bounded reward $0 \le r(x,y) \le \|r\|_\infty = 1$ which implies that $\sigma_{\alpha,r}^2(x) \le \frac{\|r\|_\infty^2}{4}$, and hence we have:

$$\sigma_{\alpha,r,\varepsilon}(x) \le \sqrt{\frac{\|r\|_\infty^2}{4} + \varepsilon}.$$

We normalize the reward so that : $\sigma_{\alpha,r,\varepsilon}(x) \le \sqrt{\frac{\|r\|_\infty^2}{4} + \varepsilon} \le 1$.

We denote GRPO advantage function as:

$$A_\alpha(x,y) = \frac{r(x,y) - \mu_{\alpha,r}(x)}{\sigma_{\alpha,r,\varepsilon}(x)}$$

$$\mathcal{L}_\alpha(\pi(\cdot|x)) = \mathbb{E}_{y\sim\alpha(\cdot|x)} \frac{\pi(y|x)}{\alpha(y|x)} A_\alpha(x,y)$$

If $\alpha = \pi_k$, we obtain the online policy objective function of GRPO, where the advantage is computed with the current policy $\pi_k$, i.e using $A_{\pi_k}(x,y)$.

We have:

$$\mathcal{L}_\alpha(\pi(\cdot|x)) = \frac{1}{\sigma_{\alpha,r,\varepsilon}(x)} \left( \mathbb{E}_{y\sim\pi(\cdot|x)} r(x,y) - \mu_{\alpha,r}(x) \right)$$

$$= \frac{1}{\sigma_{\alpha,r,\varepsilon}(x)} J(\pi(\cdot|x)) - \frac{1}{\sigma_{\alpha,r,\epsilon}(x)} J(\alpha(\cdot|x))$$

Our goal is to provide an upper bound on :

$$\mathcal{L}_\alpha(\pi(\cdot|x)) - (J(\pi(\cdot|x)) - J(\pi_k(\cdot|x)))$$

Hence we have:

$$\mathcal{L}_\alpha(\pi(\cdot|x)) - (J(\pi(\cdot|x)) - J(\pi_k(\cdot|x))) = \left( \frac{1}{\sigma_{\alpha,r,\varepsilon}(x)} - 1 \right) J(\pi(\cdot|x)) + J(\pi_k(\cdot|x)) - \frac{1}{\sigma_{\alpha,r,\varepsilon}(x)} J(\alpha(\cdot|x))$$

$$= \left( \frac{1}{\sigma_{\alpha,r,\varepsilon}(x)} - 1 \right) (J(\pi(\cdot|x)) - J(\alpha(\cdot|x)) + J(\alpha(\cdot|x))) + J(\pi_k(\cdot|x)) - \frac{1}{\sigma_{\alpha,r,\varepsilon}(x)} J(\alpha(\cdot|x))$$

$$= \frac{1 - \sigma_{\alpha,r,\varepsilon}(x)}{\sigma_{\alpha,r,\varepsilon}(x)} (J(\pi(\cdot|x)) - J(\alpha(\cdot|x))) + (J(\pi_k(\cdot|x)) - J(\alpha(\cdot|x))) + \frac{1}{\sigma_{\alpha,r,\varepsilon}(x)} J(\alpha(\cdot|x)) - \frac{1}{\sigma_{\alpha,r,\varepsilon}(x)} J(\alpha(\cdot|x))$$

$$= \frac{1 - \sigma_{\alpha,r,\varepsilon}(x)}{\sigma_{\alpha,r,\varepsilon}(x)} (J(\pi(\cdot|x)) - J(\alpha(\cdot|x))) + (J(\pi_k(\cdot|x)) - J(\alpha(\cdot|x)))$$

**Lemma 1** (Kantorovich-Rubenstein duality of total variation distance)**.** *The Kantorovich-Rubenstein duality (variational representation) of the total variation distance is as follows:*

$$\mathbb{TV}(m_1, m_2) = \frac{1}{2L} \sup_{g \in \mathcal{G}_L} \left\{ \mathbb{E}_{Z \sim m_1}[g(Z)] - \mathbb{E}_{Z \sim m_2}[g(Z)] \right\}, \tag{8}$$

*where $\mathcal{G}_L = \{g : \mathcal{Z} \to \mathbb{R}, \|g\|_\infty \le L\}$.*

On the other hand using Lemma 1 we have:

$$J(\pi(\cdot|x)) - J(\alpha(\cdot|x)) \le 2 \|r\|_\infty \mathbb{TV}(\pi(\cdot|x), \alpha(\cdot|x))$$

and

$$J(\pi_k(\cdot|x)) - J(\alpha(\cdot|x)) \le 2 \|r\|_\infty \mathbb{TV}(\pi_k(\cdot|x), \alpha(\cdot|x))$$

By our assumption on the reward and $\varepsilon$ we have :

$$\frac{1 - \sigma_{\alpha,r,\varepsilon}(x)}{\sigma_{\alpha,r,\varepsilon}(x)} \ge 0$$

so that we obtain the final bound as follows:

$$\mathcal{L}_\alpha(\pi(\cdot|x)) - (J(\pi(\cdot|x)) - J(\pi_k(\cdot|x))) \le 2 \frac{1 - \sigma_{\alpha,r,\varepsilon}(x)}{\sigma_{\alpha,r,\varepsilon}(x)} \|r\|_\infty \mathbb{TV}(\pi(\cdot|x), \alpha(\cdot|x)) + 2 \|r\|_\infty \mathbb{TV}(\pi_k(\cdot|x), \alpha(\cdot|x))$$

We obtain finally our lower bound on policy improvement as follows:

$$\boxed{J(\pi(\cdot|x)) - J(\pi_k(\cdot|x)) \ge \mathcal{L}_\alpha(\pi(\cdot|x)) - 2 \frac{1 - \sigma_{\alpha,r,\varepsilon}(x)}{\sigma_{\alpha,r,\varepsilon}(x)} \|r\|_\infty \mathbb{TV}(\pi(\cdot|x), \alpha(\cdot|x)) - 2 \|r\|_\infty \mathbb{TV}(\pi_k(\cdot|x), \alpha(\cdot|x))}$$

Integrating over $x$ (the prompts) we have:

$$\mathbb{E}_{x \sim \rho_\mathcal{X}} J(\pi(\cdot|x)) - \mathbb{E}_{x \sim \rho_\mathcal{X}} J(\pi_k(\cdot|x)) \ge \mathbb{E}_{x \sim \rho_\mathcal{X}} \mathcal{L}_\alpha(\pi(\cdot|x)) - 2 \|r\|_\infty \mathbb{E}_{x \sim \rho_\mathcal{X}} \frac{1 - \sigma_{\alpha,r,\varepsilon}(x)}{\sigma_{\alpha,r,\varepsilon}(x)} \mathbb{TV}(\pi(\cdot|x), \alpha(\cdot|x))$$

$$- 2 \|r\|_\infty \mathbb{E}_{x \sim \rho_\mathcal{X}} \mathbb{TV}(\pi_k(\cdot|x), \alpha(\cdot|x))$$

$$\ge \mathbb{E}_{x \sim \rho_\mathcal{X}} \mathcal{L}_\alpha(\pi(\cdot|x)) - 2 \|r\|_\infty \sqrt{\mathbb{E}_{x \sim \rho_\mathcal{X}} \frac{(1 - \sigma_{\alpha,r,\varepsilon}(x))^2}{\sigma_{\alpha,r,\varepsilon}^2(x)}} \sqrt{\mathbb{E}_{x \sim \rho_\mathcal{X}} \mathbb{TV}^2(\pi(\cdot|x), \alpha(\cdot|x))} - 2 \|r\|_\infty \mathbb{E}_{x \sim \rho_\mathcal{X}} \mathbb{TV}(\pi_k(\cdot|x), \alpha(\cdot|x))$$

# D   EXPERIMENTS

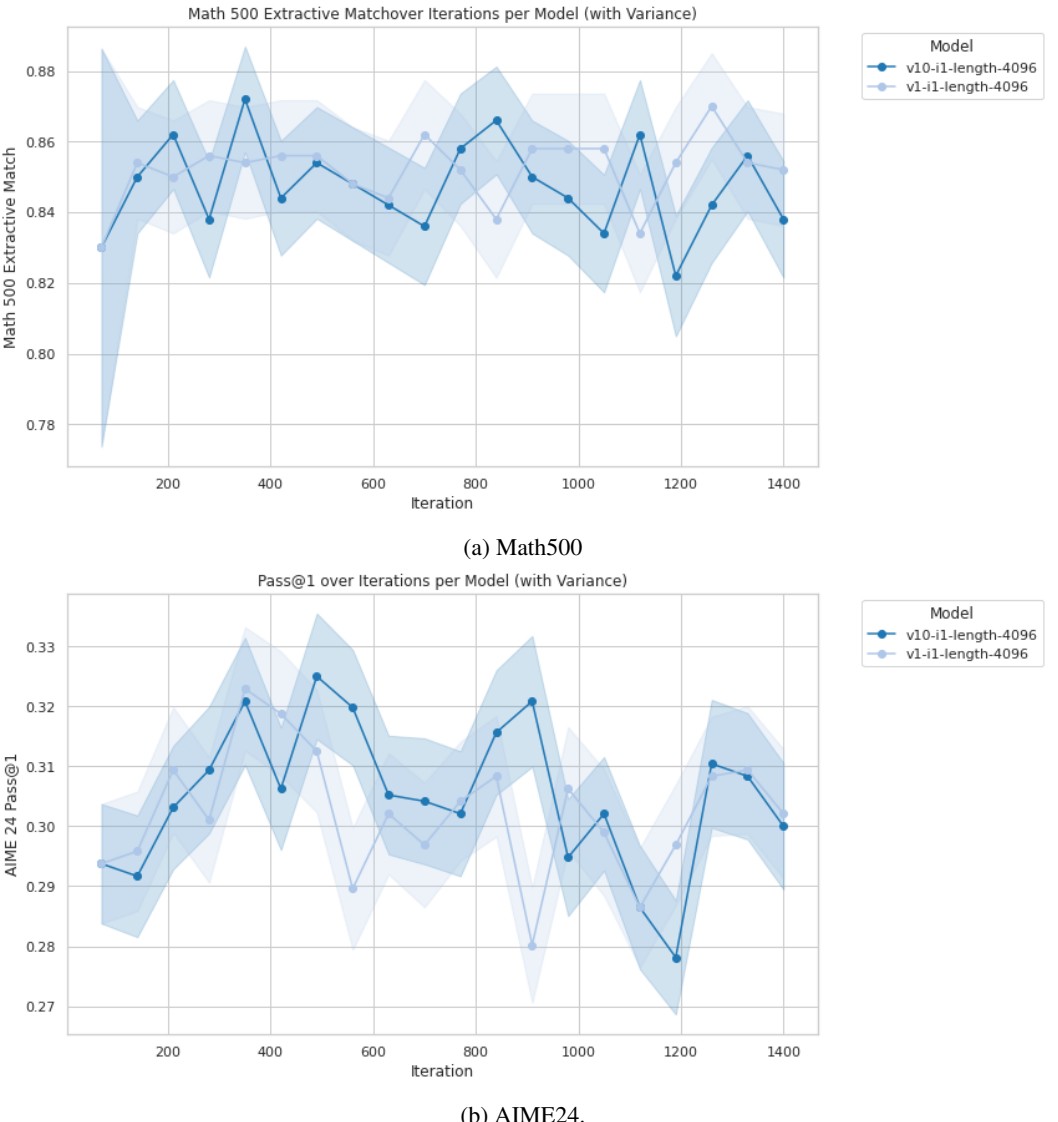

(a) Math500

(b) AIME24.

Figure 3: Aime 24/ Math 500

