# OpenReview forum: "Revisiting Group Relative Policy Optimization: Insights into On-Policy and Off-Policy Training"
_ICLR.cc/2026/Conference — ICLR 2026 Poster_

### Official Review · Reviewer_3Tjo · 2025-10-28

**Soundness:** 3
**Presentation:** 3
**Contribution:** 3
**Rating:** 6
**Confidence:** 3

**Summary:**

The paper revisits Group Relative Policy Optimization (GRPO) and develops an off-policy variant alongside a theoretical analysis of policy improvement in both on- and off-policy regimes. The authors (i) prove lower bounds showing that maximizing a GRPO-style advantage (a whitened reward) improves expected reward, for both on-policy and off-policy cases under bounded rewards and distributional closeness; (ii) derive a clipped surrogate for off-policy GRPO with a KL regularizer; and (iii) empirically compare configurations that differ in sample reuse and serving/update frequency, reporting that off-policy GRPO is at least on par and sometimes better than on-policy GRPO, with potential serving-time savings.

**Strengths:**

1.The paper extends GRPO analysis to an off-policy setting with an explicit policy-improvement lower bound, and derives an off-policy clipped surrogate analogous to PPO while exploiting GRPO’s normalized reward form.

2.For practitioners training LLMs with GRPO, the paper offers a principled path to reduce serving/communication cost in colocated/tensor-parallel settings without hurting accuracy.

**Weaknesses:**

1.The primary practical claim hinges on setting v>1, and the paper uses v=10 in its experiments. This value seems arbitrary. The theory only vaguely suggests v should not be "too large". The paper would be much stronger with an ablation study analyzing the trade-off between v, wall-clock speedup, and final task performance.

2.The improvement bound depends on total-variation terms and a variance factor (1−σ)/σ that explodes when reward variance is near zero (e.g., consistently correct/wrong prompts). The paper suggests masking such cases, but more discussion is needed on how often this occurs in practice for common verifiable rewards and what happens when masking is imperfect.

3.Most results are on math datasets with small models (0.5B / 1.5B); evidence that the method scales or competes with strong recent baselines (e.g., DAPO/DR-GRPO, off-policy PPO variants) is thin.

**Questions:**

See Weaknesses

---

> ### Author Response · Authors · 2025-11-20
> **Response to Reviewer 3Tjo**
>
> We thank the reviewer for their positive feedback and suggestions and address in the following their main questions:
> ___
>
> **1.The primary practical claim hinges on setting v>1, and the paper uses v=10 in its experiments. This value seems arbitrary. The theory only vaguely suggests v should not be "too large". The paper would be much stronger with an ablation study analyzing the trade-off between v, wall-clock speedup, and final task performance.**
>
> The choice of v indeed needs to strike a balance between being off-policy and hence computationally efficient (which tends to increase v) and not striving away too much from being on-policy (which tends towards low v). Empirically, we observed a degradation in the performance of the models for v>10, hence we chose 10, which strikes a good tradeoff for  maintaining performance while already offering good speedups.
>
> We added in Section 5.3 for the 7B model plots for the KL versus performance as function of the training iterations for on policy (i=1,v=1) and off-policy GRPO (i=1,v=10). We notice that while the KL spikes at times for off-policy GRPO, the accuracy of off-policy GRPO is not affected and it  tracks that of on-policy GRPO.
>
> ___
>
> **The improvement bound depends on total-variation terms and a variance factor (1−σ)/σ that explodes when reward variance is near zero (e.g., consistently correct/wrong prompts). The paper suggests masking such cases, but more discussion is needed on how often this occurs in practice for common verifiable rewards and what happens when masking is imperfect.**
>
> For verifiable rewards, a prompt is masked if the model produces a group of fully correct or full wrong answers. Whether this happens a lot or not depends in practice on the group size, the larger the group size, the more likely it is to encounter groups with at least a success and at least a failure.
> This mechanism results in masking introducing a bias that is tied to the group size, as large group size will cause some samples that would have been otherwise masked with small group size to be unmasked and incorporated in the loss. A solution to this was proposed in DAPO via having a dynamic group size that is gradually increased until at least a success and a failure are met, while not exceeding a given maximum group size.  We added a discussion of this in the paper in remark1.
> ___
> **3.Most results are on math datasets with small models (0.5B / 1.5B); evidence that the method scales or competes with strong recent baselines (e.g., DAPO/DR-GRPO, off-policy PPO variants) is thin.**
>
> We appreciate the Reviewer's comments about the scope of our experiments. We would like to clarify the intent of our empirical verifications. Our primary goal with the provided experiments is to provide empirical support to our theory and to validate that off-policy GRPO is competitive with a strong on-policy GRPO baseline, rather than to claim SOTA across RL methods. We view this as a foundation for future work that will more systematically compare against a broader range of approaches.
>
> Regarding model scale and baselines, we note that:
> For the 0.5 B and 1.5 B models we  directly compared on-policy and off-policy GRPO variants under controlled conditions
> We also evaluated DAPO-style masking of samples with rewards that have zero variance in experiments with the 0.5B model.
>
> In Section 5.3, we present results on the larger **Qwen2.5-7B** model using the orz-math-57k reasoning dataset, where off-policy GRPO demonstrates meaningful training speedups over on-policy GRPO.
>
> We concur with the Reviewer that a comprehensive comparison against DAPO/DR-GRPO and off-policy PPO across multiple model sizes would provide a more thorough empirical contribution. We view such an extensive empirical benchmarking effort as valuable empirical follow-ups that should build on the theoretical foundations established in this work in future investigations.

---

### Official Review · Reviewer_dUmt · 2025-10-28

**Soundness:** 3
**Presentation:** 1
**Contribution:** 2
**Rating:** 2
**Confidence:** 3

**Summary:**

This paper proposes and analyzes an off-policy variant of GRPO (a recent policy gradient algorithm for training LLMs).
In particular, the authors identify two sources that render the policy gradient estimator off-policy: the optimization loop, which increases the discrepancy between the behavior policy and the target policy, and the use of past data.

The authors test the algorithm both by training from scratch and by finetuning an existing model (Deep Seek) on math problems, showing that the off-policy version does not lose performance relative to the on-policy version.

**Strengths:**

The paper analyzes GRPO, a recent, simple, and efficient algorithm for training LLMs.
The authors, inspired by recent off-policy variants of PPO, propose an Off-Policy variant of GRPO. They also propose a policy improvement bound that covers both off-policy and on-policy development.

**Weaknesses:**

First, let me write here that I am not familiar with the LLM literature. However, I am very familiar with the reinforcement learning literature and understand GRPO and PPO well. I scored my own assessment of the paper with _confidence=3_ for this reason.

Clarity
--------

The paper's presentation, in my opinion, is quite confusing. Let me state why:

* As far as I understand, the central motivation of the paper revolves around the development of a decentralized update scheme of GRPO. This point remains unclear until page 6.
* It is not clear to me why the authors use three nested loops for their policy update scheme. In particular, I am confused about a) the outer loop and the need for the "reference policy"; and b) the need to run the optimization (lines 15--17 of algorithm 1) several times even though the policy is not used to collect data. In particular, it appears that every optimization overrides the previous one, without having a real effect on the algorithm.
* The authors call off-policy GRPO, GRPO with a number of optimization steps i>1. While I acknowledge that the gradient estimator becomes off-policy  after one policy update, the overall algorithm, according to the literature, is still considered on-policy (e.g., PPO is still considered on-policy, even if it applies several optimization steps on the same data)
* The experimental section introduces acronyms and names like GSM8K, DeepScaleR-Preview-Datasets, Aime24, "extractive matrching", light-eval, and Pass@1 without explaining what they are. Perhaps they are clear for researchers from the LLM community, but they are not from me, and I needed to google them to understand the results. A bit of explanation in the paper wouldn't have hurt.
* It is not clear why the experiment Off-Policy GRPO uses S=1, while all the other experiments use S=3.

In addition to the points made, the paper is really math-dense, making it hard to follow.

Results
----------

Given the paper's scope, it is entirely acceptable for the authors to use only three seeds for their LLM experiments. However, it would have been nice to see downscaled experiments with more seeds to see statistical significance, and hyperparameter sensitivity (i and \nu). Furthermore, I am really suspicious about how the authors used the seeds: In Section 5.1, the authors report that the three seeds give such a small standard deviation (0.003--0.03) that it is not visible in the plots. I have never seen such a little deviation, and I can't understand how it is possible, since the learning curves are pretty noisy. It suggests that the policy always selects the same actions. I can't really figure out how this is possible (unless the policy is basically deterministic, which would make it very hard to train).

Most importantly, it is not clear how the off-policy version does not really outperform the on-policy version. The off-policy version reuses data and should be more sample-efficient. GePPO shows a consistent improvement over "on-policy" PPO.

Summary
-------------

The paper does not seem ready for publication at this stage. The authors should consider writing the paper for a broader audience, clarifying details and results. In my view, the empirical section is also weak, given the low variability across results, the absence of small-scale ablation studies and hyperparameter sensitivity, and the fact that the proposed algorithms do not really improve on the on-policy version.

**Questions:**

I would ask the authors to clarify all the points above. Most importantly about:

- design choices of the algorithms (outer loop and reference policy); multiple optimization steps even though the policy is not updated
- lack of variability among seeds
- why off-policy GRPO does not outperform the on-policy version
- why the experiment Off-Policy GRPO uses S=1, while all the other experiments use S=3

---

> ### Author Response · Authors · 2025-11-20
> **Response to Reviewer dUmt 1/2**
>
> We thank the review for their feedback and address in the following their main raised points:
> ___
> **As far as I understand, the central motivation of the paper revolves around the development of a decentralized update scheme of GRPO. This point remains unclear until page 6.**
>
> We clarify that we are not proposing a decentralized scheme for GRPO in the sense of decentralized distributed optimization.  We are proposing a sampling strategy to estimate the advantage motivated from importance sampling. In computing the expected advantage for GRPO, we need to apply importance sampling: this can be done with the current policy (v=1, on- policy), or with a lag from a previous policy (v=10, off-policy).
>
> ___
>
> **Algorithm:  design choices of the algorithms (outer loop and reference policy); multiple optimization steps even though the policy is not updated
> .**
>
> We clarify here:
> After finishing a full  training epoch of GRPO, Shao et al propose to change the reference model in GRPO to the last checkpoint from the last epoch. Note GRPO’s objective consists in maximizing the advantage and in distilling the reference model. By swapping the reference  to the last checkpoint from the previous epoch, we ensure we are improving upon this new reference.
>
> The original GRPO paper of shao et al proposes indeed optimizing for several iterations without collecting new data, this is standard also in PPO training.  This style of training  was originally proposed in PPO [1], and analyzed in Section 4.1 in [2] as an off-policy training.
>
> Indeed the data is reused for i iterations, if i is small the model policy that is being updated is not far from the data providing policy and can be seen as off-policy training. The rational behind this data reuse from the PPO paper is to improve performance.
> All GRPO opensource implementations such as TRL  allow this style of data reuse in GRPO, PPO etc. For example this parameter i in TRL is `num_iterations`.
>
> [1] John Schulman, Filip Wolski, Prafulla Dhariwal, Alec Radford, and Oleg Klimov. Proximal policy optimization algorithms. arXiv preprint arXiv:1707.06347, 2017.
>  [2] Transductive Off-policy Proximal Policy Optimization
> ___
> **The authors call off-policy GRPO, GRPO with a number of optimization steps i>1 ... according to the literature, is still considered on-policy (e.g., PPO is still considered on-policy, even if it applies several optimization steps on the same data)**
>
> We followed [2] (Section 4.1) in considering this type of training as off-policy, in the sense that the “behavior” policy ($\alpha$) that provides data to the training is fixed for i iterations, while the policy being optimized ($\pi$) is changing. In addition, for small i the policy $\pi$ is not drifting a lot from the policy $\alpha$, making the "off-policy" nomenclature arguably appropriate. Our theoretical analysis, for off-policy GRPO applies also to this case as this optimization is valid as long as $TV(\pi || alpha)$ remains small.
>
> We added this discussion  in remark 1 in the updated paper.
>
> [2] Transductive Off-policy Proximal Policy Optimization
>
> ___
>
> **The experimental section introduces acronyms and names like GSM8K ... Perhaps they are clear for researchers from the LLM community, but they are not from me, and I needed to google them to understand the results**
>
> Thank you for this comment. We added explanations in the Appendix B.
> ___
> **It is not clear why the experiment Off-Policy GRPO uses S=1, while all the other experiments use S=3.**
>
> This was to ensure fairness in terms of compute time between configurations  (v=10, i=1) and  (v=1, i=1) and (v=1 , i=10). S= 3 means training for 4 epochs. The setting v=1, i=10 is the most expensive computationally, as a single epoch is 10 times slower than both (v=10, i=1) and (v=1, i=1).
> ___
>
> **Results and Statistical Significance**
>
> We confirm that we indeed used 3 seeds in the 0.5 B models experiment, and indeed we did not show the variance in the plots due to it being small.
>
> Note that we don’t do cold RL starting from random initialization, all experiments are initialized from the same reference model. As a consequence, the seed only impacts the order in which the training data is seen, as well as the sampling in the GRPO loop. At test time, for each prompt we generate 50 rollouts, each one containing a reasoning chain followed by `<answer> {integer} <answer>`. The evaluation measures only if that answer is correct or not and reports the percentage of successes averaged over the whole dataset. The lack of appreciable variability is due to 1) all models being  initialized from the same reference model, 2) the evaluation of correctness is already an average of 50 trials for each prompt, which is a standard procedure specifically in LLM papers on mathematical reasoning problems.

---

> ### Author Response · Authors · 2025-11-20
> **continued Response 2/2**
>
> ___
> **Most importantly, it is not clear how the off-policy version does not really outperform the on-policy version.**
>
> We would like to clarify that there are two types of off-policy training **(v=10, i=1)** (ours) and **(v=1, i=10)** (Shao et al). (v=10, i=1) provides fresh samples at each gradient update from a fixed anchor model for v=10 steps. And (v=1, i=10) optimizes the model for i steps using fixed data from a fixed anchor model. The reviewer seems to be referring to (v=1, i=10) as the only off-policy regime, but we should clarify as we will also do in the revisions of the paper, that our proposal is  (v=10, i=1) that is an off-policy setting with fresh samples.
>
> Our off-policy GRPO (i=1, v=10) outperforms the on-policy (i=1,v=1) on the 0.5 B models experiments. The (i=10, v=1) suffers from overfitting.
>
> On 7 B model experiments we observed that on policy and off-policy (i=1,v=10) are on par in terms of performance, with off-policy however providing **1.35x speedup**. The main goal of off-policy is indeed computational efficiency as it aims to reduce communication costs in deploying the model at each iteration to sample from it.

---

### Official Review · Reviewer_E1KW · 2025-11-01

**Soundness:** 3
**Presentation:** 3
**Contribution:** 3
**Rating:** 6
**Confidence:** 3

**Summary:**

This paper revisits **Group Relative Policy Optimization (GRPO)**, a reinforcement learning algorithm widely adopted for post-training large language models (LLMs) with verifiable rewards.
While prior work (Shao et al., 2024) introduced GRPO as an on-policy variant of PPO using standardized rewards (thus avoiding a critic network), this paper extends it to the **off-policy regime**, theoretically analyzes **reward improvement guarantees**, and empirically validates the new formulation.

The authors:

* Prove **policy improvement lower bounds** for both on-policy and off-policy GRPO (Theorem 1, Corollary 1).
* Derive a **clipped surrogate objective** analogous to PPO but specialized for GRPO’s whitened rewards.
* Provide a **masking strategy** for zero-variance samples (grounded theoretically).
* Demonstrate experimentally that **off-policy GRPO** matches or exceeds the on-policy variant while reducing **communication costs** in distributed LLM training setups.
  Experiments cover **GSM8K**, **DeepScaleR**, and scaling tests on **Qwen2.5-7B** models.

**Strengths:**

1. **Theoretical Rigor and Novelty**

   * Provides the **first formal policy improvement bound for GRPO**, including both on- and off-policy cases.
   * The proofs elegantly extend beyond standard MDP-based analyses, avoiding dependence on state visitation distributions and instead exploiting GRPO’s analytical advantage form.
   * The derivation of **clipped surrogate objectives** for off-policy GRPO generalizes the PPO framework in a principled way.

2. **Practical Impact for LLM Training**

   * Introduces a simple off-policy configuration (controlled by parameters *v* and *i*) that directly reduces communication overhead when serving models via vLLM.
   * Empirically confirms the **stability and efficiency** of this approach, a highly relevant concern for multi-GPU and TP-based deployments.

3. **Bridges Gaps Between RL Theory and LLM Post-Training**

   * Clarifies the relationship between GRPO, PPO, DAPO, and off-policy PPO variants (G-PPO, ToPPO) within a unified analytical framework.
   * Provides theoretical justification for “zero-variance masking” (previously used heuristically in DAPO) by linking it to the reward improvement bound.

4. **Clear Experimental Verification**

   * Ablation studies on GSM8K (Pass@1) demonstrate that both **masking** and **off-policy training** improve stability and accuracy.
   * Finetuning experiments on DeepSeek-R1-Distill-Qwen 1.5B confirm the off-policy variant’s ability to maintain or slightly improve performance on **AIME24** and **Math500** benchmarks.
   * Scaling experiments with **Qwen 2.5 7B** demonstrate ~1.35× speedup without loss in performance.

5. **Reproducibility and Transparency**

   * The codebase is based on public TRL and Open-R1 repositories, ensuring reproducibility.
   * Detailed setup and hardware configurations are included in the appendix.

**Weaknesses:**

1. **Limited Experimental Diversity**

   * Experiments focus primarily on math datasets (GSM8K, DeepScaleR). Broader testing on reasoning, code generation, or instruction-following datasets would strengthen generality.
   * The number of random seeds (three) is relatively small, and there’s little reporting of variance beyond Pass@1 standard deviation.

2. **Theory–Practice Gap**

   * While the proofs are strong, some constants in the policy improvement bound (e.g., the variance-dependent term (1-σ_{α,r,ε}/σ_{α,r,ε})) lack empirical interpretation or sensitivity analysis.
   * More intuition or visualization for these terms would help practitioners understand stability conditions.

3. **Presentation and Readability**

   * Dense notation (π, α, v, µ, ε, β, σ, Δ) makes the theoretical sections heavy.
   * Some equations (especially Theorem 1 and derivations in Appendix B) could benefit from concise summaries or diagrams.
   * Figures could include clearer captions and consistent scaling.

4. **Minor Empirical Limitations**

   * The speedup results (1.35×) are modest; while communication cost reductions are meaningful, scaling analysis beyond a single node would be more convincing.
   * Comparison to other RLHF or off-policy RL methods (e.g., OPPO, ToPPO) is missing in quantitative form.

**Questions:**

1. How does the choice of *v* (off-policy lag) affect both convergence and stability? Are there theoretical or empirical bounds on how large *v* can be before degradation occurs?
2. Could the policy improvement bounds be extended to **partially observable or multi-step MDPs** (where GRPO is applied iteratively on dialogue tasks)?
3. Does masking zero-variance samples introduce bias in reward estimation over time?
4. How does off-policy GRPO behave when the reward distribution is highly skewed or continuous rather than Bernoulli?

---

> ### Author Response · Authors · 2025-11-20
> **Response to Reviewer E1KW**
>
> We thank the reviewer for their  positive feedback and address their main questions:
> ___
> **How does the choice of v (off-policy lag) affect both convergence and stability? Are there theoretical or empirical bounds on how large v can be before degradation occurs?**
>
> We'd like to thank the reviewer for the suggestion. We addressed this question by adding in Section 5.3 new results on the 7B model, and plotting the KL versus performance as function of the training iterations for on policy (i=1,v=1) and off-policy GRPO (i=1,v=10). We notice that while the KL spikes at times for off-policy GRPO, the accuracy  of off-policy GRPO is not affected and it  tracks that of on-policy GRPO.
>
> The choice of v indeed has an important effect as we observed a degradation in the performance of the models for v>10. Following this observation we chose v=10 since it offers a good tradeoff in terms of maintaining performance while already offering considerable speedup.
> ___
>
> **Could the policy improvement bounds be extended to partially observable or multi-step MDPs (where GRPO is applied iteratively on dialogue tasks)?**
>
> That is an interesting question, we don’t know if the improvement bounds for GRPO can be extended to partially observed or multi-step MDPs.
> Addressing the multi-step setting will need to define a Generalized Advantage for GRPO, similarly to what was done for multi-step TRPO in [a]. Indeed [a]  shows that multi step TRPO coincides with Generalized advantage estimation. This is an interesting venue for future research.
>
> For POMDP, it is not clear how to incorporate the belief updates within the GRPO framework as the "advantage" is computed using the statistic of observed trajectories. A GRPO-like framework for POMDPs is an interesting but currently completely open research question..
> [a] Multi-step Greedy Reinforcement Learning Algorithms
> ___
>
> **Does masking zero-variance samples introduce bias in reward estimation over time?**
> For verifiable rewards, a prompt is masked if the model produces a group where the answers are either all correct or all wrong.
> In general, masking indeed introduces a bias that is tied to the group size. Enlarging the group size will make it more likely to encounter groups with at least one failure and at least one success, which would result in the unmasking (and their contribution to the loss) of samples  that could have been otherwise masked with smaller group sizes. A solution to this was proposed in DAPO via having a dynamic group size that is gradually increased until at least a success and a failure are met, while not exceeding a given maximum group size.
> In the new version of the paper we added a discussion of this bias in remark 1.
>
> ___
> **How does off-policy GRPO behave when the reward distribution is highly skewed or continuous rather than Bernoulli?**
>
> Our off-policy GRPO theory holds for any bounded rewards, including bounded   continuous rewards or bounded skewed rewards. If the reward is not bounded  one can always calibrate the reward using empirical CDFs or via a normalization as done for example in Infalign [a].
>
> [a] InfAlign: Inference-aware language model alignment

---

### Official Review · Reviewer_PeXH · 2025-11-02

**Soundness:** 3
**Presentation:** 3
**Contribution:** 2
**Rating:** 6
**Confidence:** 4

**Summary:**

The paper revisits GRPO and develops a unified view of on-policy and off-policy variants by deriving a lower bound on reward improvement. From a KL-regularized objective, the authors obtain a clipped surrogate objective for GRPO and provide a theoretical rationale for masking zero-variance samples, which would otherwise worsen the bound constants. Building on this, they propose off-policy GRPO with two practical knobs: $v$ (server/model update frequency) and $i$ (SGD steps per batch). This configuration aims to reduce communication overhead in multi-GPU/tensor-parallel setups without sacrificing accuracy. Empirically, the paper reports that on-policy can be unstable on GSM8K, masking stabilizes training, and off-policy (e.g., $v=10, i=1$) attains similar or slightly better accuracy. On larger models (7B with TP), the off-policy setting yields a reported $\sim$1.35× speedup per iteration.

**Strengths:**

- An explicit derivation of a reward-improvement lower bound for GRPO (on/off-policy) and a principled route from KL-constraint to a clipped surrogate objective.
- The $(i,v)$ design is simple and addresses a real bottleneck (communication/model updates) in multi-GPU/TP training.
- The analysis explains why zero-variance reward samples degrade constants and how masking can stabilize training in verifiable-reward tasks.
- Results on reasoning benchmarks show that the off-policy variant is competitive with the on-policy baseline while improving throughput.

**Weaknesses:**

- Most comparisons are within the GRPO family (on vs off; with/without masking). The paper should include controlled head-to-head comparisons with **DAPO / DR-GRPO / OPPO / ToPPO / DPO / IPO** under matched rewarders, sampling budgets, and compute. And the 7B setting reports system speed but lacks full accuracy curves and significance tests. Sensitivity to temperature, context length, $\beta$, group size $G$, and $(i,v)$ needs more systematic coverage.
-  Evidence is concentrated on verifiable rewards. It remains unclear how the bound and masking behave with non-binary or hard-to-verify rewards.
-  The paper states that $v$ should not be too large, but lacks an empirical guideline (e.g., KL/TV drift vs. $v$ curves) or a recommended operating range.
- Minor writing/formatting fixes for camera-ready: unify the epsilon symbol (use ε or define $\epsilon$ and use it consistently), normalize dataset names across text/tables/figures (e.g., AIME24/Math500, no spaces around the slash), standardize the tool name to a single spelling (e.g., lighteval), correct a few misspellings/grammar (libraries, standardized, laid; learning rate $5×10^6$), and tighten math displays so TV is typeset as $TV^{2}$ and long fractions/clipping expressions aren’t broken across lines.

**Questions:**

1. Can the authors add **DAPO/DR-GRPO/OPPO/ToPPO/DPO/IPO** results with identical rewarders, sampling temperature, context length, and compute budgets (ideally including a 7B+ model)?
2. Please report KL/TV drift curves across training for different $v$ and provide a recommended range for stable/efficient operation.
3. How is the zero-variance masking threshold chosen in practice? Can the authors provide a sample utilization vs. accuracy/variance?
4. Do the conclusions hold for code generation, safety alignment, or open-ended QA with graded or noisy reward signals?

---

> ### Author Response · Authors · 2025-11-20
> **Response to Reviewer  PeXH**
>
> We thank the reviewer for the careful reading and the positive assessment of our theoretical analysis, masking justification, and the practical off-policy design, and address here their main questions. Typos and minor points have been taken care of.
> ____
>
> **Can the authors add DAPO/DR-GRPO/OPPO/ToPPO/DPO/IPO results with identical rewarders, sampling temperature, context length, and compute budgets (ideally including a 7B+ model)?**
>
> Given the tight time for the rebuttal we are not able to provide all these ablations that need a lot of resources. For example an experiment on the Qwen2.5-7B model on the reasoning math dataset orz-math-57k that we provide in the paper, requires 3 days of training for a single epoch on  a single node with 8 GPUs, using trl's GRPO with vLLM- colocate and a tensor parallel size TP=4. Note that this experiment is run for beta=0 as mentioned in the paper in line 466, meaning it is in DAPO style training (that is a variant of GRPO). This choice allows for considerable memory savings (since for beta=0 we don’t need to keep the reference model in memory), without which this experiment could not be run on a single node with 8 GPUs.
>
> Despite not being able to provide all requested comparisons for 7B+ models we provide smaller scale experiments that we hope that the Reviewer will appreciate. For the 0.5 B and 1.5 B models we compared GRPO variants of on-policy and off-policy with beta>0, and we also considered for the 0.5B model DAPO style masking of samples with rewards that have zero variance. As for 7B models, as explained above we showed that for beta=0 as in DAPO, we obtain a competitive performance for off-policy on on-policy GRPO on a single node with a speedup of 1.35 for off-policy GRPO.
>
> We hope that even without all the requested additional comparisons, the Reviewer will appreciate that the experiments that we present are intended to validate the theory and verify that off-policy GRPO is competitive with a strong on-policy GRPO baseline, rather than to claim SOTA across RL methods.
> ___
>
> **Please report KL/TV drift curves across training for different  and provide a recommended range for v for stable/efficient operation.**
>
> As suggested, in Section 5.3 we added plots with the KL versus performance as function of the training iterations for on policy (i=1,v=1) and off-policy GRPO (i=1,v=10) for the 7B model. We notice that while the KL spikes at times for off-policy GRPO, the accuracy  of off-policy GRPO is not affected and it  tracks that of on-policy GRPO.
>
> We will also clarify our choice for v, based on our observation that the performance of the models for v>10  starts to degrade, and hence we choose 10 since it strikes a nice tradeoff for maintaining performance while offering speedups.
> ___
>
> **How is the zero-variance masking threshold chosen in practice? Can the authors provide a sample utilization vs. accuracy/variance?**
>
> The variance is computed within the group for each prompt, any non-zero estimated variance is not masked. Hence, we don’t use any parametric threshold in practice, as prompts with answers that are fully correct or fully wrong are masked.
>
> ___
> **Do the conclusions hold for code generation, safety alignment, or open-ended QA with graded or noisy reward signals?**
>
> GRPO has been demonstrated to be effective across use cases such as code generation with execution rewards that are binary, a setting similar to ours. However, similarly to discussed references like [c], we chose to limit our experiments to mathematical reasoning tasks, which are currently the most established benchmarks for reasoning using RL with verifiable rewards.
> Regarding other types of rewards such as safety alignment and open-ended QA, the  presented theory needs only bounded rewards which can always be achieved via some normalization or calibration of the reward as in the InfAlign paper. We expect that our conclusions will hold in this case as well.
>
> Regarding the robustness of GRPO to noise, we believe this is tied to the performance of the reference on the task and the signal to noise ratio between distilling the reference model and in how large the noise is as reported in the recent paper [a].
>
>
> [a] Mirage or Method? How Model-Task Alignment Induces Divergent RL Conclusions
> [b] InfAlign: Inference-aware language model alignment
> [c] DAPO: An Open-Source LLM Reinforcement Learning System at Scale

---

### Meta-Review · Area_Chair_A6Nx · 2025-12-24

**Summary:**

This paper studies the GRPO algorithm, which is an on-policy RL algorithm in LLM training. The authors extend the GRPO algorithm to the off-policy setting and analyze the reward improvement guarantees. Advantageous experimental performance are shown for the off-policy GRPO method.

Although the authors have not addressed all the concerns from the reviewers, they have provided meaningful justifications. The only negative feedback that makes this paper borderline is from reviewer dUmt. By carefully reading the complaints of the reviewer as well as the authors' responses. The AC believes that the authors have addressed the concerns from this reviewer (where the concerns are mostly due to unfamiliarity to the topic). If the reviewer given the chance to adjust the score, the AC thinks the reviewer would change the rating. Therefore, the AC decide to accept this paper.

**Reviewer Concerns:**

Addressed or justified:
(1) PeXH & E1KW: a few concerns on compared benchmarks and experiment settings.
(2) 3Tjo: more results on larger models.
(3) dUmt: writing clarity issues.

**Reviewer Scores:**

We note that the only reviewer with negative rating is dUmt. After carefully reading the complaints and the authors' responses, the AC believes that the concerns are mostly due to unfamiliarity to the topic, the authors have addressed the concerns from this reviewer. If the reviewer were given the chance to adjust the score, she/he will possibly rise the score to above the acceptance threshold.

---

### Decision · Program_Chairs · 2026-01-26

Accept (Poster)